# Fat mass loss correlates with faster disease progression in amyotrophic lateral sclerosis patients: Exploring the utility of dual-energy x-ray absorptiometry in a prospective study

Ikjae Lee[1]*, Mohamed Kazamel[1], Tarrant McPherson[2], Jeremy McAdam[3], Marcas Bamman[3,4,5], Amy Amara[1], Daniel L. Smith, Jr.[6], Peter H. King[1,3,5]*

1 Department of Neurology, University of Alabama at Birmingham, Birmingham, Alabama, United States of America, 2 Department of Biostatistics, University of Alabama at Birmingham, Birmingham, Alabama, United States of America, 3 Department of Cell, Developmental, & Integrative Biology, University of Alabama at Birmingham, Birmingham, Alabama, United States of America, 4 Department of Medicine, University of Alabama at Birmingham, Birmingham, Alabama, United States of America, 5 Birmingham Veterans Affairs Medical Center, Birmingham, Alabama, United States of America, 6 Department of Nutrition Sciences, University of Alabama at Birmingham, Birmingham, Alabama, United States of America

* IL2384@cumc.columbia.edu (IL); phking@uabmc.edu (PHK)

**Data Availability Statement:** All relevant data are within the manuscript and its Supporting information files.

## Abstract

### Background/objective

Weight loss is a predictor of shorter survival in amyotrophic lateral sclerosis (ALS). We performed serial measures of body composition using Dual-energy X-ray Absorptiometry (DEXA) in ALS patients to explore its utility as a biomarker of disease progression.

### Methods

DEXA data were obtained from participants with ALS (enrollment, at 6- and 12- months follow ups) and Parkinson's disease (enrollment and at 4-month follow up) as a comparator group. Body mass index, total lean mass index, appendicular lean mass index, total fat mass index, and percentage body fat at enrollment were compared between the ALS and PD cohorts and age-matched normative data obtained from the National Health and Nutrition Examination Survey database. Estimated monthly changes of body composition measures in the ALS cohort were compared to those of the PD cohort and were correlated with disease progression measured by the Revised Amyotrophic Lateral Sclerosis Functional Rating Scale (ALSFRS-R).

### Results

The ALS cohort (N = 20) had lower baseline total and appendicular lean mass indices compared to the PD cohort (N = 20) and general population. Loss in total and appendicular lean masses were found to be significantly associated with follow-up time. Low baseline percentage body fat (r = 0.72, p = 0.04), loss of percentage body fat (r = 0.81, p = 0.01), and total fat

**Funding:** This work was supported by the National Institute of Neurological Disorders and Stroke (www.nih.org), R01NS092651and R21NS111275-01(PHK), and by the Dept. of Veterans Affairs (www.va.gov), BX001148 (PHK). Sponsors and funders had no role in study design, data collection and analysis, decision to publish, or preparation of the manuscript.

**Competing interests:** The authors have declared that no competing interests exist.

mass index (r = 0.73, p = 0.04) during follow up correlated significantly with monthly decline of ALSFRS-R scores in ALS cohort who had 2 or more follow-ups (N = 8).

## Conclusion

Measurement of body composition with DEXA might serve as a biomarker for rapid disease progression in ALS.

## Introduction

Amyotrophic lateral sclerosis (ALS) is a heterogenous disorder defined by degeneration of upper and lower motor neurons leading to progressive motor dysfunction and muscle atrophy. Death typically occurs 3 to 5 years after symptom onset and is most often related to respiratory failure. Weight loss is common in ALS patients during the course of disease due to direct muscle loss, decreased caloric intake, and increased metabolic demand [1]. Rapid weight loss has been recognized as an indicator of faster progression and shorter survival [2, 3]. There is an ongoing effort to address weight loss by supplying high caloric nutrition as a therapeutic intervention; however, the benefit of this approach has not been clearly demonstrated [4–6].

A current limitation is the paucity of studies assessing the underpinnings of weight loss in ALS patients, including the contributions of lean and fat mass loss in disease progression. Multiple factors affect weight and nutritional status in ALS. Degeneration of lower motor neurons leads to denervation and ultimately loss of muscle fibers resulting in a decrease in muscle mass. Energy metabolism is altered in ALS patients and generally the energy requirement increases despite reduced muscle mass, mobility and other motor activity [7–9]. Nutritional and caloric intake are frequently decreased due to loss of appetite, dysphagia, and difficulty in feeding [10, 11]. While all these factors can cause negative caloric balance and weight loss, the degree of loss in fat and muscle is likely variable among individual patients with ALS. Adding to this complexity, the asymmetrical nature of ALS can lead to variable changes in body segments and sides. Therefore, measuring simple weight does not reflect the important distinctions of body composition as a whole or for specific body regions.

Whole-body Dual-Energy X-Ray Absorptiometry (DEXA) reliably measures body components including bone, fat and lean masses [12]. Few studies have followed ALS patients longitudinally using DEXA scans [5, 13], and they are limited by the number of subjects and follow-up period. In this study, we compared body composition data obtained by DEXA scans of ALS patients to that of the general United States population (age- and sex-matched). We included a cohort of Parkinson's disease (PD) patients as an additional control. This neurodegenerative disease is associated with progressive motor deficits of a different type and underlying pathology. We further analyzed longitudinal follow-up data from DEXA body composition analyses and correlated these with disease progression and survival in a subset of the ALS cohort.

## Methods

### Subjects

The study protocol was approved by the University of Alabama at Birmingham (UAB) and Birmingham VA Medical Center (BVAMC) Institutional Review Boards (IRB). ALS participants were recruited from UAB and BVAMC between June 2016 and September 2019 after informed consent. Inclusion criteria included patients who met El Escorial criteria of possible,

probable or definite ALS and were 21 years or older. Patients with moderate to severe neuropathy and pregnant women were excluded. Demographics, height, weight, premorbid weight, and disease-related history were collected at enrollment. Participants were followed prospectively for a maximum of 12 months. Revised Amyotrophic Lateral Sclerosis Functional Rating Scale (ALSFRS-R) scores were obtained by designated and trained study coordinators at enrollment and every month. A change of ALSFRS-R (ΔALSFRS-R) per month was calculated by subtracting the enrollment score from the last follow up score and divided by the duration. For patients who expired during the study period, monthly change of ALSFRS-R was calculated by using ALSFRS-R as 0 at the deceased date. Whole body DEXA scans were obtained at enrollment (baseline), 6-month, and 12-month follow ups. Data for the PD cohort were obtained from a no-exercise control group in our recently completed, randomized controlled trial of persons with PD [14]. All PD subjects underwent whole body DEXA scan at enrollment and at 4 months.

## Measurement of body composition

Height and weight were measured for each participant and recorded to the nearest millimeter and 0.1kg respectively. Each participant was scanned with a whole-body DEXA scan (GE-Lunar iDXA, Madison, WI) using the manufacturer's guidelines and were analyzed using enCORE 2008 version 12.3 software. Scan times lasted approximately 5–10 minutes. Body composition variables including fat mass, lean tissue mass, bone mineral content, and bone mineral density were determined using the GE Lunar enCORE software standard analysis modules. In addition to total body composition, regional estimates were made for the arms, legs and trunk. Appendicular lean tissue mass was calculated by adding lean tissue mass of the arms, and legs. Among all the body composition parameters, total fat mass, total tissue % fat (%Fat), total lean mass, and appendicular lean mass were of primary interest in this analysis. Body mass index (BMI) was calculated by the following formula: weight(kg)/height(m)$^2$. Total Lean Mass Index (TLMI), Appendicular Lean Mass Index (ALMI), and Total Fat Mass Index (TFMI) are also height-adjusted indices. Changes in body composition variables were calculated by subtracting the enrollment value from the follow up value for mixed models comparing ALS and PD cohorts over time. Monthly changes in BMI and body composition variables (ΔBMI, ΔTLMI, ΔALMI, ΔTFMI, Δ%Fat) were calculated by subtracting the enrollment value from the last follow up value and divided by duration (in months) between scans in order to examine their correlation with changes per month of ALSFRS-R.

## Statistical analysis

Demographics including age, sex, and race were described and compared between ALS and PD cohorts using Student's t-test for continuous variables and Fisher's exact test for categorial variables. Height, weight, BMI, TLMI, ALMI, TFMI, and %Fat were described and compared between ALS and PD cohorts using Student's t-test. TLMI, ALMI, TFMI, and %Fat from ALS and PD cohorts were compared to the age, sex and race matched general US population and categorized based on percentile ranges of the general population: < 3rd percentile, 3rd to < 10th percentile, 10th to < 50th percentile, 50th to < 90th percentile, 90th to < 97th percentile, and > 97th percentile by using National Health and Nutrition Examination Survey (NHANES) provided DEXA reference data obtained from over 20,000 adults and children between 1999–2004 in the United States [15]. The z scores for baseline TLMI, ALMI, TFMI, and %Fat were calculated for the ALS and PD cohorts based on the LMS method and NHANES reference data [15] and compared to the general United States population by using one sample Wilcoxon signed-rank tests and to each other using Wilcoxon rank-sum tests.

Patient characteristics at enrollment (age, sex, disease duration, dysphagia, dyspnea, ALSFRS-R, pre-enrollment change of ALSFRS-R per month, weight, premorbid weight, BMI, premorbid BMI, % weight change from premorbid to enrollment) of ALS cohort who had 2 or more DEXA scans were described separately. Associations between changes in body composition and disease group, follow up interval and baseline body composition were analyzed by using univariable and multivariable mixed model with compound symmetric covariance to account for multiple measurements within subjects and Kenward-Roger approximations [16] to denominator degrees of freedom in Wald test statistics. Correlations between monthly change of ALSFRS-R with monthly change in body compositions, and baseline body compositions were examined using Pearson Correlation analysis. Statistical significance was defined at p<0.05 and because of the exploratory nature of this paper, no adjustments for multiple comparisons were done. Analyses were performed using SAS version 9.4 and programs from the R project version 3. 3. 2.

## Results

A total of 20 ALS subjects were enrolled in the study, with 65% being male and 75% Caucasian. Mean age at enrollment was 59 (range 35–81). At the beginning of the study, 90% of recruited subjects met El Escorial criteria of probable or definite ALS with 10% being possible ALS. By the end of the study period, all subjects met criteria for probable or definite ALS. Seventy-five percent of the subjects had spinal onset and 25% had bulbar onset disease, 65% had dysphagia and 65% had dyspnea at enrollment. Median disease duration at enrollment was 23 months (Interquartile range 11–32). Seventy percent of the subjects were taking riluzole while 40% were receiving edaravone. The average ALSFRS-R score at enrollment was 30 (SD +/- 9.9). A PD cohort (n = 20 total) was selected from a randomized, controlled trial (see Methods) and matched for sex and aligned closely by age with the ALS cohort. The median disease duration in the PD cohort was 2 years and all patients were ambulatory. Disease severity at enrollment was mild in 90% and moderate in 10% based on Hoehn & Yahr stage (stages 2–3) and MDS-UPDRS scores [17]. Dysphagia was absent or minimal in 90%, significant in 5%, and unknown in 5% based on UPDRS 2.3 sub-score. The proportion of Caucasians in the PD cohort was higher than in the ALS cohort, although this difference was not statistically significant [Table 1]. Average height was similar between the two cohorts, while weight and BMI trended lower in the ALS cohort. TLMI and ALMI were significantly lower in ALS cohort (p = 0.002 and 0.008 respectively). TFMI tended to be lower while %Fat appeared higher in the ALS cohort, but neither reached statistical significance [Table 1].

**Table 1. Demographic and anthropometric characteristics of the study groups.**

|  | ALS (20) | PD (20) | p-value |
|---|---|---|---|
| Sex, % male | 65% | 65% | 1 |
| Age at Enrollment, mean (SD) | 59.4 (10.8) | 63.0 (3.8) | 0.2 |
| Race, % White % Black | 75%, 25% | 90%, 10% | 0.4 |
| Height, cm, mean (SD) | 174.8 (8.2) | 173.5 (9.9) | 0.7 |
| Weight, kg, mean (SD) | 78.1 (22.7) | 84.9 (17.7) | 0.28 |
| BMI, mean (SD) | 25.3 (5.8) | 28.2 (4.5) | 0.09 |
| Total Lean Mass Index, mean (SD) | 15.1 (2.8) | 17.7 (2.1) | 0.002 |
| Appendicular Lean Mass Index, mean (SD) | 6.9 (1.7) | 8.2 (1.3) | 0.008 |
| Total Fat Mass Index, mean (SD) | 9.1 (3.8) | 9.4 (3.4) | 0.8 |
| Percentage fat, mean% (SD) | 36.7 (6.7) | 33.9 (7.9) | 0.2 |

TLMI, ALMI, TFMI, and %Fat measurements in individual subjects of the ALS and PD cohorts were compared to an age-, sex- and race-matched general population obtained from the NHANES database as described in the methods. The ALS cohort had a significantly lower baseline TLMI and ALMI compared to the general population (p = 0.0001, p = 0.004 respectively) [Fig 1A and 1B]. The PD cohort had a significantly higher baseline ALMI compared to the general population (p = 0.006). TFMI and %Fat were not significantly different between either cohort and the general population [Fig 1C and 1D].

Among 20 ALS subjects, 8 had two or more DEXA scans and were included for further analysis. Demographics and disease characteristics are described in Table 2. Univariable and multivariable mixed model analysis using change of body composition indices as an outcome and baseline body composition index, months between DEXA scans and disease group as variables were performed to examine factors potentially related to changes in body composition indices. Change in ALMI was significantly associated with both disease groups (p<0.01) and follow up duration (p = 0.0005) when tested individually. When disease group, follow up duration and baseline ALMI were included in the multivariable model, only follow up duration was significantly associated with the change of ALMI (p = 0.02) while the disease group was not (p = 0.8). Similarly, change in TLMI was significantly associated with disease group (p<0.05) and follow up duration (p = 0.003) in an univariable model while only follow up duration (p = 0.003) was significantly associated in multivariable model with disease group and baseline TLMI. In an univariable mixed model, changes of TFMI, %Fat were not significantly associated with baseline indices, disease group or follow up duration [Table 3].

Regular assessment of ALSFRS-R, TLMI, ALMI, TFMI and %Fat were plotted for each of the participants during the study period [Fig 2A–2E]. Based on the change in ALSFRS-R per month, 5 participants (A02, A03, A04, A07, A08) had a slow to intermediate progression (rate of decline less than 1.2 points per month) and 3 participants (A01, A05, A06) had rapid progression (rate of decline more than 1.2 points per month). A01 had spinal onset disease with progression rate of 2.2 prior to study enrollment. A05 had bulbar onset disease with progression rate of 2.1 prior to study enrollment. A06 had a progression rate of 1.1 prior to study enrollment, however, had bulbar onset disease and low forced vital capacity at enrollment. All 3 subjects had dysphagia at enrollment. Among these 3 subjects with rapid progression, 2 died during the study period and the other died 4 months after study completion due to respiratory insufficiency despite non-invasive positive pressure ventilation. Fig 2F demonstrates the striking change of body silhouette on serial DEXA scanning in participant A01 who was a fast progressor.

Change per month (Δ) in ALSFRS-R scores were correlated with baseline and change per month in BMI, TLMI, ALMI, TFMI, %Fat. ΔALSFRS-R correlated significantly with ΔTFMI, Δ%Fat, and baseline %Fat [Fig 3A–3C]. The correlation coefficient was highest with Δ%Fat (r = 0.81). Correlation between ΔALSFRS-R and ΔBMI, ΔTLMI, ΔALMI, BMI, TLMI, ALMI, TFMI did not reach statistical significance. [Fig 3D–3F and S1 Fig] Similarly, ΔALSFRS-R scores correlated significantly with ΔTFMI, Δ%Fat, and baseline %Fat but not with ΔBMI, ΔTLMI, ΔALMI, BMI, TLMI, ALMI, TFMI when last ALSFRS-R follow up scores were used instead of using ALSFRS-R score of 0 at the time of death for those who deceased during the study period.

## Discussion

Weight change is closely linked to ALS progression and may precede onset of weakness by decades.(1–6, 23) Thus, weight preservation may serve as a target of disease treatment if we understand the pathophysiology behind it. Considering that weight is composed of different

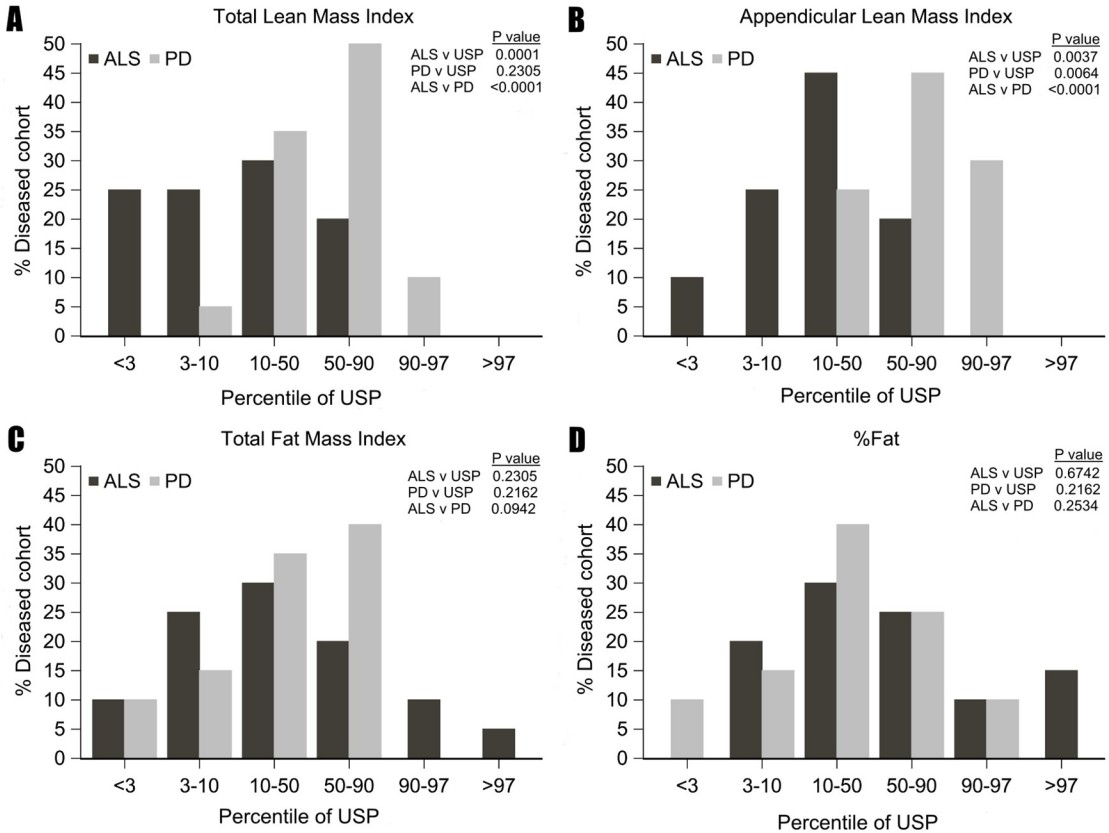

**Fig 1. Percentile distribution of ALS and PD cohorts at enrollment based on z scores compared to age, sex and race matched United States Population (USP).** The ALS cohort had significantly lower total lean mass index (A) and appendicular lean mass index (B) compared to the PD cohort and USP. The PD cohort had significantly higher appendicular lean mass compared to USP (B). Total fat mass index (C) and percent fat (D) were not different between ALS and PD cohorts and USP. P-values are derived from Wilcoxon signed-rank test of ALS cohort vs USP, PD cohort vs USP, and from Wilcoxon rank-sum test of ALS cohort vs PD cohort.

body components, including lean and fat mass, longitudinal measurement of these components might provide further insight into determinants of ALS disease progression. In this study, we showed that change in total fat mass and body fat percentage correlated significantly with rapid disease progression while change in BMI, total and appendicular lean masses did not. These findings indicate that the measurements of fat mass with DEXA might serve as a sensitive imaging biomarker for disease progression. Baseline body fat percentage also correlated with rapid disease progression raising its potential as a prognostic biomarker. We have demonstrated that the ALS cohort had lower total and appendicular lean mass compared to age- and sex-matched PD patients and to the United States population at enrollment while no significant differences were noted with fat mass and body fat percentage. Total and appendicular lean masses declined significantly over the course of the study, while fat mass and body fat percentage change did not correlate with follow up duration. Our results collectively suggest that lean mass loss is almost universal in all patients due to ALS disease progression while fat mass is associated with rapid progression of disease. These results are consistent with the previous studies that showed lean mass invariably declines but fat mass either increases or decreases depending on the stage of disease and energy balance [13, 18]. The results from our study are also in line with prior observations that adiposity correlates with prognosis in ALS,

**Table 2. Characteristics of ALS participants with 2 or more DEXA scans.**

| Subjects | Sex | Age | Disease duration (mos) | Onset | Dysphagia | Dyspnea | Initial ALSFRS-R | ΔALSFRS at enrollment† | Initial forced vital capacity | Riluzole | Edaravone | Weight (kg) Premorbid* | Weight (kg) Enrollment | BMI Premorbid* | BMI Enrollment* |
|---|---|---|---|---|---|---|---|---|---|---|---|---|---|---|---|
| A01 | M | 42 | 9 | Spinal | Y | Y | 28 | 2.2 | 68% | N | N | 102.1 | 94.8 | 30.5 | 28.3 |
| A02 | M | 56 | 29 | Spinal | N | Y | 34 | 0.5 | 75% | N | Y | 94.0 | 96.8 | 29.7 | 30.6 |
| A03 | F | 45 | 25 | Spinal | N | N | 44 | 0.2 | 75% | Y | Y | 64.0 | 60.3 | 24.2 | 22.8 |
| A04 | M | 67 | 16 | Spinal | N | N | 44 | 0.3 | 67% | Y | N | 123.2 | 119.3 | 37.9 | 36.7 |
| A05 | M | 59 | 9 | Bulbar | Y | Y | 29 | 2.1 | 65% | Y | N | 70.5 | 68.9 | 23.0 | 22.4 |
| A06 | M | 64 | 8 | Bulbar | Y | N | 39 | 1.1 | 20% | Y | N | 80.8 | 74.3 | 27.9 | 25.7 |
| A07 | M | 45 | 16 | Spinal | Y | Y | 30 | 1.1 | 91% | N | N | 108.8 | 98.2 | 33.0 | 29.8 |
| A08 | M | 63 | 18 | Spinal | Y | Y | 29 | 1.1 | 95% | Y | Y | 136.5 | 127.0 | 38.6 | 35.9 |

*Premorbid weight is defined by patient reported stable weight at least 1 year prior to symptom onset

† ΔALSFRS at enrollment was calculated by the following formula: (48-initial ALSFRS-R)/Disease duration)

**Table 3. Multivariable mixed model analysis of change in body compositions and baseline body compositions, disease group and follow up duration.**

| | Effect | | | |
|---|---|---|---|---|
| **Outcome** | **Intercept** | **Baseline Body Composition Index** | **Group (PD reference)** | **Follow Up Duration** |
| Change in TLMI | -0.3205 | 0.05662 | 0.2744 | -0.1716** |
| | | | | (p = 0.003) |
| Change in ALMI | -0.1378 | 0.0574 | -0.0613 | -0.0835* |
| | | | | (p = 0.02) |
| Change in TFMI | -0.0239 | 0.0607 | -0.0163 | 0.0004 |
| Change in %Fat | -0.4149 | 0.0644 | -0.4186 | -0.0657 |

BMI, body mass index; TLMI, total lean mass index; ALMI, appendicular lean mass index; TFMI, total fat mass index; %Fat, percentage body fat

with higher adiposity associated with longer survival and lower adiposity with shorter survival [19–25].

Dysphagia and inadequate nutrition have been recognized as independent and modifiable risk factors for faster disease progression and shorter survival in ALS [26], leading to a number of prospective studies and randomized trials attempting to address these factors [4–6]. A most recent randomized double-blinded placebo-controlled trial assessing a high-caloric, fatty diet demonstrated that high-caloric supplements did not have a survival benefit in their ALS cohort as a whole [6]. Post hoc analysis, however, showed a favorable response in the fast progressing subgroup. Although further study will be required for definitive conclusions, these results underscore the importance of identifying subsets of ALS patients that might benefit from early nutritional intervention. Our study suggests that measurement of body composition with DEXA might serve as an imaging biomarker to identify patients with faster disease progression who could benefit from aggressive nutritional intervention including pharmacological treatment. This subset would not be identified by the standard ALSFRS-R which focuses on functional impairment.

DEXA is currently considered a gold standard to measure body compartments and to provide quantitative values for total and segmental fat mass, lean mass, fat mass ratio, bone mineral density, and bone mineral content [15, 27]. It is noteworthy that lean mass from DEXA is not equal to muscle mass as it also includes water, organ and other non-fat non-bone soft tissue [28]. BMI has been widely used to estimate fatness; however, it has been shown to be inaccurate in ALS [29]. Our study also demonstrated that the change in BMI did not correlate significantly with disease progression as opposed to direct measurement of fat mass. Air Displacement Plethysmography tends to overestimate body fat in thinner participants and underestimate body fat in obese patients [30]. DEXA was overall well tolerated in this cohort of ALS patients. However, it does require the patient to lay flat which might be difficult for ALS patients with advanced respiratory insufficiency. Bioelectrical impedance analysis (BIA) measurement is comparable to DEXA in ALS patients [31], and thus can be considered as an alternative in measuring body composition.

While the PD data were derived from a separate study, the DEXA scans were obtained by the same protocol, equipment and technical staff which encouraged us to use them as a disease control. PD is a neurodegenerative disorder with progressive motor deficits including tremor, bradykinesia and rigidity with an entirely different underlying pathology. Compared to the ALS cohort, lean mass and fat mass stayed relatively stable in the PD cohort, however, the observation period was shorter in the PD cohort. In a study by Yong et al, DEXA-measured

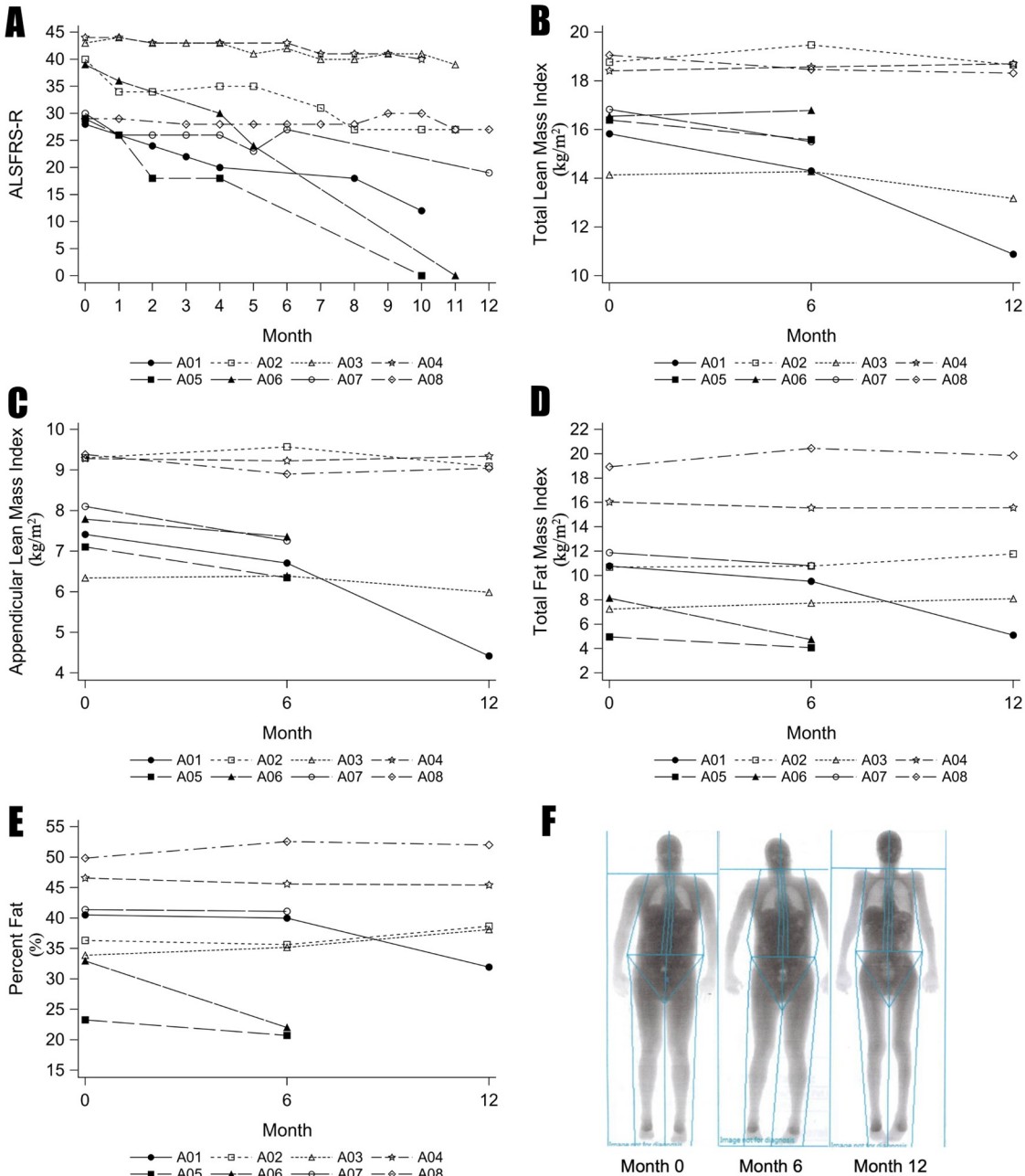

**Fig 2. Baseline and change in DEXA measurements in ALS patients with two or more scans over the study period.** (A) ALSFRS-R, (B) Total Lean Mass Index, (C) Appendicular Lean Mass Index, (D) Total Fat Mass Index, and (E)) Percent Fat. Fast progressors are indicated with filled symbols and intermediate to slow progressors are indicated with open symbols. (F) Serial DEXA scans of patient A01 who was a fast progressor.

body fat and lean masses did decline in PD patients when followed for 3 years suggesting that the decline in body mass loss does occur over a longer timeframe [32].

There are several limitations to our study. First the sample size was small although representative of the demographics and clinical features of larger groups of ALS patients previously reported by our group and others [33, 34]. Second, follow up studies were hampered by patient

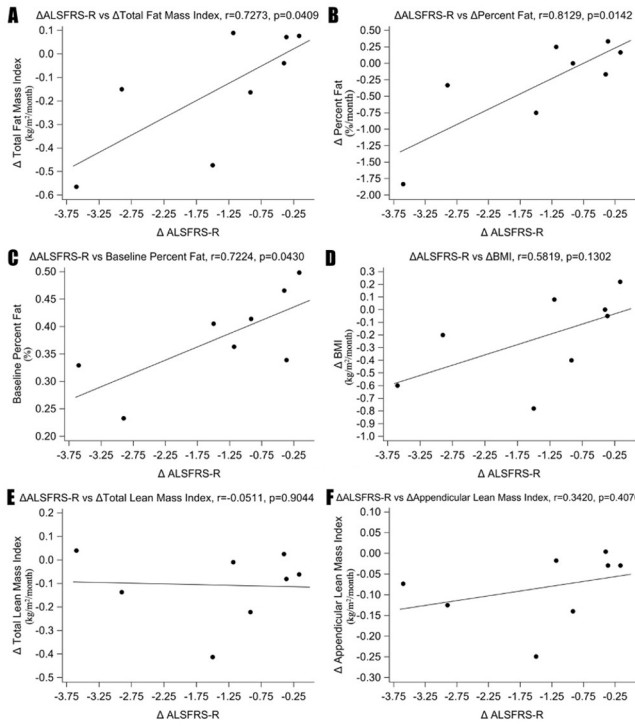

**Fig 3. Correlation between ΔALSFRS-R with baseline and longitudinal DEXA measurements.** ΔALSFRS-R correlates significantly with (A) ΔTotal Fat Mass Index, (B) ΔPercent Fat and (C) baseline percent fat. The correlation is not significant with (D) ΔBMI, (E) ΔTotal Lean Mass Index and (F) ΔAppendicular Lean Mass Index.

drop out related to such factors as poor mobility, transportation challenges, and death. Despite these limitations, we were still able to detect significant shifts in body composition and this will help focus future validation studies with a larger sample size. Additional measurements of appetite, dietary intake, and activity level might help to determine the most salient contributors to fat mass change and the balance between energy intake and expenditure in ALS patients in future studies. Ultimately, a randomized, controlled, clinical trial will be needed to determine whether nutritional intervention that targets fat mass preservation/maintenance can slow down disease progression and prolong survival in ALS.

## Supporting information

**S1 Fig. Insignificant correlation between ΔALSFRS-R with baseline and longitudinal DEXA measurements.** ΔALSFRS-R and (A) Baseline Total Fat Mass Index, (B) Baseline BMI, (C) Baseline Total Lean Mass Index and (D) Baseline Appendicular Lean Mass Index are not significant.
(TIF)

## Acknowledgments

We would like to thank the UAB Nutrition Obesity Research Center's Metabolism Core (P30DK056336, Core Director: Dr. Barbara Gower; Robert Petri) for assistance with DEXA measures.

## Author Contributions

**Conceptualization:** Ikjae Lee, Mohamed Kazamel, Daniel L. Smith, Jr., Peter H. King.

**Data curation:** Ikjae Lee, Jeremy McAdam, Marcas Bamman, Amy Amara, Peter H. King.

**Formal analysis:** Ikjae Lee, Tarrant McPherson, Peter H. King.

**Funding acquisition:** Peter H. King.

**Investigation:** Ikjae Lee, Mohamed Kazamel, Peter H. King.

**Methodology:** Ikjae Lee, Jeremy McAdam, Peter H. King.

**Project administration:** Ikjae Lee, Mohamed Kazamel, Amy Amara, Peter H. King.

**Resources:** Marcas Bamman, Amy Amara, Daniel L. Smith, Jr., Peter H. King.

**Software:** Peter H. King.

**Supervision:** Ikjae Lee, Marcas Bamman, Peter H. King.

**Validation:** Ikjae Lee, Peter H. King.

**Visualization:** Ikjae Lee, Tarrant McPherson, Peter H. King.

**Writing – original draft:** Ikjae Lee.

**Writing – review & editing:** Ikjae Lee, Mohamed Kazamel, Tarrant McPherson, Jeremy McAdam, Marcas Bamman, Amy Amara, Daniel L. Smith, Jr., Peter H. King.

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
