## [Decision Letter · Decision Letter 0]

18 Mar 2021

PONE-D-20-36763

Fat mass loss correlates with faster disease progression in amyotrophic lateral sclerosis patients: exploring the utility of Dual-Energy X-ray Absorptiometry in a prospective study

PLOS ONE

Dear Dr. Lee,

Thank you for submitting your manuscript to PLOS ONE. After careful consideration, we feel that it has merit but does not fully meet PLOS ONE’s publication criteria as it currently stands. Therefore, we invite you to submit a revised version of the manuscript that addresses the points raised during the review process.

Please address the relatively minor issues raised by the reviewers and then the paper can be considered for publication.

We look forward to receiving your revised manuscript.

Kind regards,

Matti Douglas Allen, PhD

Academic Editor

PLOS ONE

Journal Requirements:

Reviewers' comments:

Reviewer's Responses to Questions

**Comments to the Author**

1. Is the manuscript technically sound, and do the data support the conclusions?

Reviewer #1: Yes

Reviewer #2: Partly

2. Has the statistical analysis been performed appropriately and rigorously? 

Reviewer #1: Yes

Reviewer #2: Yes

3. Have the authors made all data underlying the findings in their manuscript fully available?

Reviewer #1: Yes

Reviewer #2: Yes

4. Is the manuscript presented in an intelligible fashion and written in standard English?

Reviewer #1: Yes

Reviewer #2: Yes

5. Review Comments to the Author

Reviewer #1: Lee et al present an interesting study on the utility of DEXA scans as a progression biomarker in ALS using a small cohort of ALS subjects. This is compared to national data as well as a cohort of PD subjects using the same DEXA protocol, but different follow up times. They show in a longitudinal subset of the ALS subjects that DEXA markers do associate with ALS progression via ALSFRS-R. These findings are interesting, although the biggest limitation of this study is the very small sample size, with only 8 subjects in the longitudinal analysis. Overall the work is sound and well presented. The limitations are stated.

Abstract:

-Suggest including more details including the sample sizes and findings on DEXA (i.e. present the statistics). The current abstract is very generic and doesn't reflect what turns out to be an interesting paper.

Paper:

-Authors mention that neuropathy is exclusionary, why? How many individuals did this exclude? How many subjects were approached for this study and declined?

-Were study coordinators trained in ALSFRS-R assessment?

-What is the value of including the PD subjects? The follow up durations are different and clinically, there often is not a clinical question of whether a DEXA 2 years into ALS disease is needed to differentiate ALS from PD. In other words, the DEXA would not be needed at a diagnostic biomarker at this disease stage. To me, especially given the follow up DEXA differences in the fast and slow groups, these data could be removed with the focus on the paper on the case only analysis (plus a NHANES comparison). This focuses and makes the paper more interesting in my opinion.

Overall, nice paper that meets PLOS ONE acceptance standards.

Reviewer #2: In this study, the authors measured body composition of subjects with ALS and compared changes in Lean and Fat mass body composition percentages to disease progression, measured by ALSFRS-R. Baseline body composition values at study onset were compared to known values of the general population and subjects with Parkinson's Disease. The primary finding of the study was that ALS subjects had decreased total and appendix lean mass at study onset and disease progression correlated with loss of total fat and fat mass percentage. Therefore, tracking changes in ALS fat mass likely corresponds with disease progression.

A major criticism for this manuscript is how the authors anticipate utilization of study results, as a disease biomarker. To compare subject fat loss with disease progression, the authors used a less cumbersome method to establish disease severity, the ALSFRS-R. Given that baseline measurement of neither %Fat nor TFMI predicted disease progression, it isn't clear how using longitudinal DEXA measurements is more efficient or accurate at showing disease progression than longitudinal ALSFRS-R scores alone. Therefore, much more explanation is required as to how DEXA is to be incorporated in an additive way into patient care to derive meaningful information that can't be obtained with simpler, already established methods.

This isn't to say that the study results don't advance scientific knowledge in the field of ALS, it is a comment of the positioning of the information as a 'biomarker' that is problematic. As the authors noted, clinical studies investigating use of dense caloric diets in ALS have shown mixed results, perhaps due to disease severity. As noted in a 2011 Lancet paper, metabolic abnormalities appear to exist in a subset of ALS patients, with some speculation that disease severity may be associated with increased metabolism and potentially a metabolic switch from OXPHOS to glycolysis, which is less efficient in glucose utilization. Your data seems to support this hypothesis, showing that a subset of patients, likely correlating with those of increased disease severity, is hypermetabolic and increasing nutritional calories may delay but not change the disease progression.

Minor comments:

In the abstract: as only 8 people completed the study, it seems like a more descriptive statistic here would be how many subjects you were able to use data from. You could say, 20 people enrolled, or something else if you want to keep the number 20 in the abstract, but using an N=20 seems inaccurate given your data set in the results.

The use of PD as a control needs to be explained more. Yes, PD is a neurodegenerative disease, but it is quite different from ALS and not typically associated with decreased weight. It's not clear how PD is in any way relevant to your study aims.

Your paper mentions two references that have also used DEXA in ALS patients. Given that this study is also limited by small number of subjects and follow-up time, it's unclear how this study adds to current knowledge.

When analyzing data, it's problematic to assume equal changes over multi-month interval duration, which isn't necessarily accurate. For example, in some instances, ALSFRS-R was not measured for 4 or more months, yet a straight line was assumed between points. Have any previous studies shown that declines in ALSFRS-R tend to be consistent across time or are there rapid drop-offs at disease end-stages?

6. PLOS authors have the option to publish the peer review history of their article (what does this mean?). If published, this will include your full peer review and any attached files.

Reviewer #1: No

Reviewer #2: No

---

## [Author Response · Author response to Decision Letter 0]

27 Mar 2021

Response to Reviewers

Comments to the Author

Reviewer #1: Lee et al present an interesting study on the utility of DEXA scans as a progression biomarker in ALS using a small cohort of ALS subjects. This is compared to national data as well as a cohort of PD subjects using the same DEXA protocol, but different follow up times. They show in a longitudinal subset of the ALS subjects that DEXA markers do associate with ALS progression via ALSFRS-R. These findings are interesting, although the biggest limitation of this study is the very small sample size, with only 8 subjects in the longitudinal analysis. Overall the work is sound and well presented. The limitations are stated.

->Thank you for your thoughtful summary and comments.

Abstract:

-Suggest including more details including the sample sizes and findings on DEXA (i.e. present the statistics). The current abstract is very generic and doesn't reflect what turns out to be an interesting paper.

->We appreciate this suggestion. We have now added the sample sizes and statistics for the correlation analysis which is the key findings in this study. Other statistics were not added in the abstract due to word count limit.

Paper:

-Authors mention that neuropathy is exclusionary, why? How many individuals did this exclude? How many subjects were approached for this study and declined?

->The study participants were recruited for a broader study to examine the muscle derived biomarkers. Therefore, ALS patients with significant neuropathy were not recruited due to potential confounding effects on the muscle. However, we do not believe any patient was excluded due to this criterion. 

-Were study coordinators trained in ALSFRS-R assessment?

->We had a dedicated study coordinator who was trained and consistently obtained ALSFRS-R throughout this study. 

-What is the value of including the PD subjects? The follow up durations are different and clinically, there often is not a clinical question of whether a DEXA 2 years into ALS disease is needed to differentiate ALS from PD. In other words, the DEXA would not be needed at a diagnostic biomarker at this disease stage. To me, especially given the follow up DEXA differences in the fast and slow groups, these data could be removed with the focus on the paper on the case only analysis (plus a NHANES comparison). This focuses and makes the paper more interesting in my opinion.

->The intention of adding the Parkinson’s subjects is to have a neurodegenerative disease control as a positive control comparator. Parkinson’s disease is a relevant control as these patients typically have progressive motor deficits but with a totally different underlying pathology. By using Parkinson’s subjects as a control, we wanted to contrast the striking changes in fat and lean mass among ALS patients that are not only different from the normal population but also from a disease with motor impairment. Furthermore, we did not have longitudinal data on the normal population, thus decided to use data from the longitudinal follow up of Parkinson’s subjects, noting the limitation due to a different follow up duration. 

We have now included this rationale for using Parkinson’s cohort in the introduction of the paper 

 

Overall, nice paper that meets PLOS ONE acceptance standards.

Reviewer #2: In this study, the authors measured body composition of subjects with ALS and compared changes in Lean and Fat mass body composition percentages to disease progression, measured by ALSFRS-R. Baseline body composition values at study onset were compared to known values of the general population and subjects with Parkinson's Disease. The primary finding of the study was that ALS subjects had decreased total and appendix lean mass at study onset and disease progression correlated with loss of total fat and fat mass percentage. Therefore, tracking changes in ALS fat mass likely corresponds with disease progression.

A major criticism for this manuscript is how the authors anticipate utilization of study results, as a disease biomarker. To compare subject fat loss with disease progression, the authors used a less cumbersome method to establish disease severity, the ALSFRS-R. Given that baseline measurement of neither %Fat nor TFMI predicted disease progression, it isn't clear how using longitudinal DEXA measurements is more efficient or accurate at showing disease progression than longitudinal ALSFRS-R scores alone. Therefore, much more explanation is required as to how DEXA is to be incorporated in an additive way into patient care to derive meaningful information that can't be obtained with simpler, already established methods.

->We do understand your concern regarding the utility of DEXA as a marker of disease progression when ALSFRS-R is easier to obtain and more directly associated with functional impairment. Our study did show that the DEXA measurement of fat mass percentage at baseline also correlated with disease progression which can be a useful prognostic marker at an earlier stage of disease. Furthermore, DEXA is an objective measurement that is not dependent on the examiners or patient reporting which are well documented limitations of ALSFRS-R. Lastly, we have emphasized that loss of fat mass correlates strongest with the fast progression. By utilizing DEXA, we might be able to identify this subgroup of patients who might require and benefit from aggressive nutritional intervention. Therefore, we believe that DEXA can be considered as an imaging biomarker with added values. We have modified the discussion section to reflect these points. 

This isn't to say that the study results don't advance scientific knowledge in the field of ALS, it is a comment of the positioning of the information as a 'biomarker' that is problematic. As the authors noted, clinical studies investigating use of dense caloric diets in ALS have shown mixed results, perhaps due to disease severity. As noted in a 2011 Lancet paper, metabolic abnormalities appear to exist in a subset of ALS patients, with some speculation that disease severity may be associated with increased metabolism and potentially a metabolic switch from OXPHOS to glycolysis, which is less efficient in glucose utilization. Your data seems to support this hypothesis, showing that a subset of patients, likely correlating with those of increased disease severity, is hypermetabolic and increasing nutritional calories may delay but not change the disease progression.

->We agree with your comments. In addition to our response to your helpful first comment, we also emphasized in the Discussion section the potential utility of identifying this “hypermetabolic” subset of patients by DEXA scanning. 

Minor comments:

In the abstract: as only 8 people completed the study, it seems like a more descriptive statistic here would be how many subjects you were able to use data from. You could say, 20 people enrolled, or something else if you want to keep the number 20 in the abstract, but using an N=20 seems inaccurate given your data set in the results.

->We have modified the abstract to clarify that the numbers of participants at baseline and follow ups. 

The use of PD as a control needs to be explained more. Yes, PD is a neurodegenerative disease, but it is quite different from ALS and not typically associated with decreased weight. It's not clear how PD is in any way relevant to your study aims.

->Please see our response to the same comment made by reviewer number 1. We have included a brief rationale in the introduction of the paper.

Your paper mentions two references that have also used DEXA in ALS patients. Given that this study is also limited by small number of subjects and follow-up time, it's unclear how this study adds to current knowledge.

->Thank you for pointing out. We believe that the main advantage of the current study is longer follow up periods up to 12 months where other studies had 6 months (Nau et al) and 4 months (Wills et al) follow up for DEXA scans. Also, previous studies have not examined the change of body composition in relation to the rate of disease progression which we did in our study. 

When analyzing data, it's problematic to assume equal changes over multi-month interval duration, which isn't necessarily accurate. For example, in some instances, ALSFRS-R was not measured for 4 or more months, yet a straight line was assumed between points. Have any previous studies shown that declines in ALSFRS-R tend to be consistent across time or are there rapid drop-offs at disease end-stages?

->A previous study has demonstrated that the trajectory of ALSFRS-R scores is variable among ALS patients and that the decline is more curvilinear than linear (PMID: 26205535). That being said, the decline of ALSFRS-R scores in our ALS cohort seems quite linear based on their measurements. Moreover, the linear projection of ALSFRS-R scores fit well with the time of death in fast progressing patients (A01, A05, A06). Therefore, we believe that the monthly decline of ALSFRS-R is a good representation for the rate of disease progression in our cohort.

---

## [Editor Report · Decision Letter 1]

20 Apr 2021

Fat mass loss correlates with faster disease progression in amyotrophic lateral sclerosis patients: exploring the utility of Dual-Energy X-ray Absorptiometry in a prospective study

PONE-D-20-36763R1

Dear Dr. Lee,

We’re pleased to inform you that your manuscript has been judged scientifically suitable for publication and will be formally accepted for publication once it meets all outstanding technical requirements.

Kind regards,

Matti Douglas Allen, MD, PhD

Academic Editor

PLOS ONE
---

## [Editor Report · Acceptance letter]

26 Apr 2021

PONE-D-20-36763R1 

Fat mass loss correlates with faster disease progression in amyotrophic lateral sclerosis patients: exploring the utility of Dual-Energy X-ray Absorptiometry in a prospective study 

Dear Dr. Lee:

I'm pleased to inform you that your manuscript has been deemed suitable for publication in PLOS ONE. Congratulations! Your manuscript is now with our production department. 

Kind regards, 

on behalf of

Dr. Matti Douglas Allen 

Academic Editor

PLOS ONE